

# The importance of better mapping of stream networks using high resolution digital elevation models – upscaling from watershed scale to regional and national scales

Anneli M. Ågren[1], William Lidberg[1],

[1] Swedish University of Agricultural Science, Department Forest Ecology and Management, S-90183 Umeå, Sweden
*Correspondence to*: William Lidberg (william.lidberg@slu.se)

**Abstract.** Headwaters make up the majority of any given stream network, yet, they are poorly mapped. A solution to this is to model the stream networks from a high resolution digital elevation model. Selecting the correct stream initiation threshold is key, but how do you do that on a national scale across physiographic regions? Here the Swedish landscape is used as a test bench to investigate how the mapping of small stream channels (<6 m width) can be improved. The best modelled stream

15 channel network was generated by pre-processing the DEM, calculating the accumulated flow, extracting a stream network using a stream initiation threshold of 2 ha. The Matthews Correlation Coefficient (MCC) for the 2 ha stream channel network was 0.463 while the best available maps of today, the Swedish property map (1:12 500) had an MCC of 0.387. A residual analysis of the 2 ha network show that there is additional improvements to be made by adapting the model to local conditions, as 15% of the over and underestimation could be explained by the variability in runoff, quaternary deposits, local topography

20 and location. The most accurate stream channel network had a length 4.5 times longer the currently mapped stream network, demonstrating how important accurate stream networks is for upscaling aquatic and climate research.



## 1 Introduction

Headwaters dominate surface water drainage networks and studies of headwater catchments have provided understanding of different sources and controls on biogeochemistry in streams (Seibert et al., 2009; Tiwari et al., 2017), and also interactions

with the atmosphere (Natchimuthu et al., 2017; Wallin et al., 2013). Despite the importance of headwaters, it's a paradox that at the same time small headwaters can be called Aqua Incognita (Bishop et al., 2008; Kuglerová et al., 2017), a relevant term as the majority of headwater streams are poorly mapped. Maps have traditionally been constructed from aerial photos, but small stream channels are difficult to observe from the air, especially under a dense tree canopy. Hence there is a bias on maps that show larger streams, and streams in agricultural areas where trees are scarce, while smaller streams are missing. Even the

best available map for Sweden (the 1:12 500 property map), will therefore severely underestimate the total length of stream networks. This can have repercussions for research questions such as process understanding and budgets. For example, in the Krycklan Study Catchment (Laudon et al., 2013) it was shown that $CO_2$ evasions was high from small scale streams, 72% of the $CO_2$ was evaded to the atmosphere from 1st to 2nd order streams (Wallin et al., 2013). Wallin et al. (2018) highlighted the importance of low order headwater streams by estimating $CO_2$ emissions from a 400 000 km stream network that was based

on a 50*50 m digital elevation model. They concluded that low order headwater streams emitted $CO_2$ corresponding to 21% of the estimated terrestrial C sequestration. Benstead and Leigh (2012) estimated that global estimates of $CO_2$ evasion from streams would increase from 0.56 Pg C $yr^{-1}$ to 1.6 Pg C $yr^{-1}$ if one assumes a 50% increase in area of small rivers (1st to 5th order).

Headwater streams have a high amount of stream edge relative to stream surface area (Richardson and Danehy 2007) and local ground water discharge is a major contributor to water flow in these streams (Leach et al., 2017). Therefore headwater streams and their riparian zones constitutes an interface between water and soil, and largely control inputs from surrounding landscapes to downstream ecosystems (Lidman et al., 2017). They also provide important ecosystem services such as cycling nutrients (Blackburn et al., 2017; Bormann and Likens, 1967) and buffering impact of pollutants (Klaminder et al., 2006). Headwater

streams also function as important habitat for both invertebrates and plants and functions as migration corridors (Freeman et al., 2007). This strong coupling between soils and streams brings up important issues regarding management of small streams. Any perturbation in headwaters will affect downstream environments, so in order to protect large streams, it's important to protect the headwater streams (Kuglerová et al., 2017; Wipfli et al., 2007).

However, in order to give small streams good protection, the first step is to know where they are. Topographical modelling using high resolution digital elevation models (DEM) derived from airborne light detection and ranging (LiDAR) is a popular





approach to map small streams (Goulden et al., 2014) and has been suggested as a solution to this problem (Ågren et al., 2014). Some advantages with this approach is that modelled stream networks form integrated drainage networks (Vaze and Teng, 2007) and follow channel depressions (Murphy et al., 2008b). There are however some problems with this approach. First, high resolution DEMs contains artificial features such as roads and require pre-processing to become hydrologically correct

(Lindsay and Creed, 2005). Secondly is to define appropriate stream initiation thresholds. This threshold is the minimum catchment area required for groundwater to become surface water and form a stream. This threshold regulates how extensive a stream network is and often require area specific knowledge or field data to make educated guesses. There are a number of studies where stream heads (Ågren et al., 2015; Avcioglu et al., 2017; Imaizumi et al., 2010; McNamara et al., 2006) or entire stream networks (Benstead and Leigh, 2002; Jensen et al. 2017; Murphy et al., 2008b) have been field mapped to validate

DEM derived stream networks. However, Jensen et al. (2017) determined that it requires a full day to field map the stream network of a 40-45 ha catchment. A research question that remains is therfore; How does one select the correct stream initiation threshold and validate DEM derived stream networks when scaling up from a catchment scale to regional or national scale? Stream initiation thresholds vary between physiographic regions (Avcioglu et al., 2017; Heine et al., 2004) and as the list of National LiDAR datasets is growing (Guo et al., 2017; van Leeuwen and Nieuwenhuis, 2010) and high resolution DEMs are

becoming accessible to managers it is important to evaluate the performance of these topographically modelled stream networks over larger scales. Therefore the aim of this study is to determine the optimum threshold for stream initiation on a national scale, using the Swedish landscape at a test bench. As to our knowledge, this is the first attempt at working with DEM derived stream networks on this large scale. We also asses how accurate these DEM derived stream channels perform in comparison to existing maps of streams and the potential to improve performance of DEM derived stream channel networks

by incorporating variability in local topography, soil texture and runoff.

## 2 Materials and methods

### 2.1 Study site

Sweden is located between latitude 55° and 70° N and longitude 11° and 25° E, placing most of the country within the boreal zone. Annual mean air temperatures range from 8 ˚C in the south to -2 ˚C in the north (Seekell et al., 2014). The bedrock is

25 mainly made up from Precambrian crystalline rocks with remains of younger sedimentary rock in the Caledonian mountains (Swedish geological institute). Sweden has been through multiple glaciations during the last 2-3 million years and most of the quaternary deposits were formed during and after the most recent glaciation, around 10 000 years ago. As a result 75 % of Sweden is covered by glacial till and 13 % is covered by peat (Fransson, 2018). The remaining 12 % consists of exposed rock, glaciofluvial and post glacial sedimentary deposits. According to the Swedish Land Cover database (based on satellite imagery)

(Ansén, 2004) the land cover in Sweden is: Forest 63.0 %, lakes 8.9 %, open mire 8.7 %,  heathlands 7.7 %, arable land 6.1 %, forested mire 2.8 %, urban areas 2.3 %, other 0.5 %.





## 2.2 Field data

As it's not possible to field map all stream channels or stream heads when working on a regional or national scale, we choose to utilise data from the Swedish national inventory of landscapes (NILS) (Ståhl et al., 2011) to evaluate the performance of modelled stream channels. The NILS inventory consists of 631 5*5 km squares, systematically distributed throughout Sweden,

covering all landscapes (forest, agricultural areas, mountains, wetlands, shores, and cities). This inventory was designed to secure statistically accurate estimates for the country as a whole and capture variability in rare landscapes. Therefore rare landscapes were sampled with a denser grid while common landscapes were sampled with sparser (Fig.1). Data collected from 631 square shaped line inventories with 200 m line segments, 12 in each square, in total 7572 segments, giving a total length of the line inventories of 1 512 km was used in this study. 3323 stream channels narrower than 6 meters were mapped in the

line inventory. Instead of mapping stream heads or entire networks, the pixels in the high resolution DEM containing a line inventory can be viewed upon as a point measurement with a presence or absence of a (small <6 m) stream channel, giving a total of 619 767 observations. Due to the national coverage of the field inventory it was impossible to investigate all stream channels during similar flow conditions. Therefore flow conditions during this inventory ranged from "temporarily dried out" to "extreme high flow". Another issue with this dataset was uncertainties in the GPS positioning. Some mapped stream

channels were up to 20 m away from its actual location. Especially underneath dense forest canopy. To account for the uncertainties in the GPS positioning the mapped stream channel points were moved, snapped, to the closest modelled stream channel within 20 m.

## 2.3 Topographically modelled stream channels

The Swedish National DEM generated by the Swedish Mapping, Cadastral and Land Registration Authority using LiDAR

data was used for hydrological modelling. The resolution was 2*2 m and was generated from a point cloud with a point-density of 0.5-1 points /$m^2$ with a horizontal and vertical error of 0.1 m and 0.3 m, respectively. The mountain region in north-western Sweden were not covered by this DEM at the time of this study. The DEM was split into 2818 sub catchments where each catchment had 2 km overlap with surrounding catchments to avoid edge effects during stream extraction. The raster stream network grids were later mosaicked back together to form a cohesive stream network before analysis. Before any hydrological

calculations could be conducted on the DEM it had to be pre-processed to become hydrologically correct (Marks et al., 1984). Lidberg et al. (2017) showed that breaching was the best way to pre-process high resolution DEMs in the Swedish landscape. Therefore a three step breaching approach was used as described below:

- Stream lines from the 1:12 500 scale property map, produced by the Swedish Mapping, Cadastral and Land

Registration Authority, were burned 1 m into the DEM on agricultural land using the tool "burn streams" in Whitebox GAT 3.4 (Lindsay, 2014).





- Stream lines from the property map were burned across road lines from the property map using the tool "burn streams at roads" in Whitebox GAT 3.4 as described by (Lidberg et al., 2017).

- Remaining sinks were resolved by the complete breaching algorithm developed by (Lindsay, 2015) using Whitebox tools (Lindsay, 2018).

A flow accumulation grid was created from the hydrologically correct DEM using deterministic-8 (D8) (O'Callaghan and Mark, 1984), a computationally efficient algorithm, that has been shown to generate accurate results on high resolution DEMs (Leach et al., 2017). Stream channel networks were extracted using stream initiation thresholds of 1 ha, 2 ha, 5 ha, 10 ha, 15 ha and 30 ha. Since the field data only contain channels <6 m, the larger streams were erased using a mask from the Swedish

Property map, where all streams >6 m were mapped as a polygon layer. Total lengths of the stream channel networks were calculated (Table 1).

## 2.4 Evaluation of accuracy

As it's not possible to field map all stream channels in Sweden, and we know from experience that the stream networks on current maps (the propery map; 1:12 500) are lacking and therefore cannot be used for validation, field data from the NILS

inventory were used to evaluate the performance of the different DEM derived stream networks. The results were evaluated using confusion matrixes, accuracy (ACC) and Mathews Correlation Coefficient (MCC) (Boughorbel et al., 2017). A confusion matrix consist of true positives (TP), i.e. where the map accurately predicts a channel. False positives (FP), i.e. where the map inaccurately predicts a channel. True negatives (TN), where the map accurately predicts the absence of a stream channel and false negatives (FN) where the map misses an existing stream channel (Fig. 2).

The confusion matrix for the modelled stream channel networks was calculated as follows:

- True positives (TP) were calculated by snapping the coordinates of the field mapped channels to the highest flow accumulation along the inventory line within 20 m. TP, is the number of observations with an accumulated flow above the selected flow initiation thresholds.

- False negatives (FN) were calculated by snapping the coordinates of the field mapped channels to the flow accumulation using a snapping distance of 20 m. FN, is the number of observations with an accumulated flow below the selected flow initiation thresholds.

- False positives (FP) were calculated as number of cells intersecting the line inventories, with a flow initiation above the thresholds, minus the number of TP. A large number of FP's was noted for the modelled stream networks. A

30 common occurrence was that a modelled stream runs along the lines of the field inventory, resulting in several adjacent cells being marked as many FP even though it's just one stream observation. To try and correct for this all



adjacent FP's were merged to one observation before calculating the FP. However when a modelled stream is meandering back and forth across the line, it is still considered as several FP.

- True negatives (TN) was calculated as the total number of cells intersecting the line inventories, minus TP, FP and FN. Due to the projection (Sweref 99 TM) the inventory lines are somewhat out of alignment from a straight N-S or W-E line, thereby introducing extra cells (each line should in theory be 1 200 cells, but the lines rage 1 222 to 1 231 cells. This introduces of an overestimation of TN of around 2%, which is negligable for the purpose of this study.

To compare the DEM derived stream networks with the current best available map the confusion matrix from the Swedish Propery map (1:12 500) was also calulated.

## 2.4 Residual analysis and potential for improvement

Residuals from the confussion matrix from the optimal stream channel network were converted into an ordinal variable; 1 for FN and 2 for FP to be used to analyse potential improvements of the stream channel network modelling. These residual points were used to extract existing map data as described below. The aim was to understand what caused over estimations (FP) and under estimations (FN) of stream channels.

Variability in runoff from between sites can affect the challelization of water. Annual runoff increases, roughly one order of magnitude, from the SE coast to the NW mountain range. Therefore runoff data provided by the Swedish metrological and hydrological institute was included. A model known as S-HYPE was used to model seasonal runoff in 33605 sub-catchments between 1982 and 2015 (Arheimer et al., 2011). Meteorological seasons of winter, spring, summer and autumn, were used to calculate seasonal as well as annual runoff for each site. Spatial differences in soils can affect permeability and drainage capacity on each site which may impact stream initiation thresholds. Therefore quaternary deposits were extracted from maps created by the Swedish Geological Survey. The scale of these maps ranges from 1:25 000 (1.7 %), 1:50 000 (2.7 %) 1:100 000 (47 %), 1:200 000 (1.4 %), 1:250 000 (21.2 %), 1:750 000 (33.6 %) and 1:1 000 000 (100 %). Some of these maps have significant overlap but and the map with the highest resolution was selected if more than one were avaliable for a site. The quartenary deposits maps were simplified into 5 classes; till, peat, rock outcrops, coarse sediment (e.g. silt to boulders), and fine sediment (e.g. clays). Local topography may also affect the channel initiation (Avcioglu et al., 2017; Imaizumi et al., 2010). Therefore local topography was calculated as a standard deviation of the digital elevation model in a moving window of 5, 10, 20, 40, and 80 cells. High values represent steep terrain and low values flat terrain. The coordinates of each site were also included in the residual analysis in order to capture other potential spatial gradients. First the X-matrix was tested for multicolinearity, using IBM SPSS Statistics 24 and since many of the explanatory variables in the residual analysis showed multicolinearity a multivariate approach was used. In order to enchance group seperation we choose to perform a Orthogonal



Projections to Latent Structures Discriminant Analysis (OPLS-DA) using SIMCA 14. Prior to analysis a balanced dataset was created by randomly selecting and equal number of FN and FP and all variables were scaled and centered.

## 3 Results

The result of the field investigation showed that stream discharge varied with time during the field investigation, 1 % were recorded during extreme high flow, 16 % during high flow, 54 % during normal flow, 11 % during low flow and 18 % channels were temporarily dried out. 33 % were natural channels, 4 % were straightened channels and ditches made up 63 %. Out of the ditches, 15 % were found in agricultural areas, 28 % were roadside ditches, the majority of the rest were found in forested land where 45 % were found in mineral soils and 11 % in peat.

### 3.1 Confusion matrix

As the flow initiation threshold decreases, the stream network expands and more of the missing headwaters are captured, as indicated by the increasing number of true positives (Table 1, Fig. 2). However, simultaneously the number of false positives also increases (Table 1, Fig. 2). As an example, out of the 3323 channels in the field inventory, only 618 were found on the property map, while the 1 ha stream channel network located 2185 channels (Table 1). A downside to the new DEM derived 1 ha stream channel network is that it also created a large number of false positives, the number of false positives increased from 143 on the Swedish property map to 5076 for the 1 ha stream network. All stream channel networks have extremely high accuracy at around 99%. However, the numbers for accuracy cannot be trusted due to the imbalance in the data (Daskalaki et al., 2006). This occurs when the sample size in the data classes are unevenly distributed. In our case most of the field sites consist of land and a typical channel intersection only occupy a single cell. For unbalanced datasets, such as this, the best measure of model performance is MCC (Boughorbel et al., 2017; Daskalaki et al., 2006). MCC for the newly derived stream networks from the DEM was highest for the 2 ha flow initiation threshold channel network (0.463).

The length of all streams under <6 m on the propertymap is 693 042 km which would corespond to a stream initation threshold of almost 30 ha (Table 1). The 2 ha network which was the most accurate was 4.5 times longer (2 639 163 km) than the stream channel network currently on the property map.

### 3.2 Residual analysis

In an ideal world, the two classes (FP and FN) would have separated on either side of the score scatter plot (Fig. 3A), however, the scores showed a large overlap between the two groups; FP and FN. The predictive power of the model was also quite low



with R2Y(cum) = 0.16 and R2X(cum) = 0.59 and Q2(cum) = 0.15, meaning that only 15 % of the variability in the X-variables was correlated to the two classes. However, despite low predictive power, we can still analyse the loading plot to detect in what landscapes FP and FN are over-represented.

According to the loading scatter plot (Fig. 3B) more FN are found in regions with high runoff during winter and in areas with high standard deviation (i.e. in steep terrain). More FP's are found in regions with high Y and X-coordinates, where summer and spring runoff is high. When looking at quaternary deposits FNs were mostly found on fine sediments while FPs were found on peat and till.

## 4 Discussion

A striking result of this study was that 81.5 % of the NILS field mapped channels were missing on the Swedish property map (Table 1). Similar result were found in a detailed study of the 68 km$^2$ Krycklan Catchment in Northern Sweden, where the stream network on current maps were missing 58 % of the perennial and 76 % of the fully expanded network during high flow (Ågren et al., 2015). MCC (Table 1) suggest that a modelled stream channel network with a flow initiation threshold of 2 ha is the most accurate stream network for the Swedish landscape, and therefore an improvement compared to the property

map. However, a downside to the new DEM derived 2 ha stream channel network is that it also created a large number of false positives, the number of false positives increased from 143 on the property map to 3291 for the 2 ha stream channel network.

The relatively low optimal channel initiation threshold of 2 ha in this study can be explained by relatively humid climate and high hydraulic conductivity of the Fennoscandia till soils but also extensive human impacts on drainage systems. It's close to

20 the 1.7 ha average channel initiation threshold found in Kansans (Heine et al. 2004). 63 % of all field mapped channels in this study were man made ditches, 4 % were straightened channels (for timer floating) and only 33 % were natural channels. Ditches have been dug for different purposes, for example; to increase rational agricultural production (Avilés et al., 2018), to increase timber production on wet forest soils or mires (Lõhmus et al., 2015) or to stabilize roads (Kalantari and Folkeson, 2013). Similar numbers were found in a study on the ditch network in the Krycklan Catchment in Northern Sweden, which

showed that the ditch network doubled the length of the stream channel network (Hasselquist et al., 2017). This is due to a long a history of ditching in the Nordic countries that stared in the late 1800s or early 1900 (Hasselquist et al., 2017). This practice has fundamentally changed the hydrology of the landscape where many patches of wet soils with subsurface flow have been ditched, draining the land, lowering the water table and creating many more channels in the landscape. Without these man-made channels the optimum flow initiation for the Swedish landscape would likely be higher. When applying the



same methodology to other landscapes and biomes it's therefore necessary to adapt the models and find the optimum flow initiation threshold for that landscape.

There is a considerable seasonal variability in the stream networks. This variability in the stream networks were for example seen in the field inventory where 18 % were recorded as temporarily dried out at the time of the investigation. A detailed study conducted in the Krycklan Catchment, Northern Sweden where the position of stream heads were field mapped during different times of the year showed that the natural stream heads were normally found at a threshold of around 10-15 ha during baseflow and expanded to 2 ha in natural streams during high-flow conditions and up to as low as 1 ha for ditches (Ågren et al., 2015). How large this inter-annual variability is largely depend on soils and climate of the site. Still, the study in the Krycklan Catchment gives conservative estimate that the length of the stream network at high-flow conditions is 2.4 times the length at baseflow conditions. Similar results was found by (Jensen et al., 2017) in the Appalachia Mountains. The length of the stream network on the property map was of the same order as the 30 ha stream network (Table 1), which would correspond to the stream network during extreme dry conditions. While this constitutes a perennial stream network it's important to realize that perennial stream networks are extreme cases and not very representative of "normal" or "average" stream networks in a landscape.

In this study a fixed threshold for the channel heads were chosen. This will introduce a bias, not only because of the seasonal variability (as discussed above) but also because the upslope areas of channel heads vary between different physiographic regions. Avcioglu et al. (2017) report that the upslope areas for channel heads range 0.03-7.6 ha in 3 physiographic regions in Alabama and Jensen et al. (2017) report correpsonding numbers of 0.3-3.9 ha in 4 physiographic regions of the Appalchian Highlands. There are other methods to determine position of channel heads in watersheds, that are based on finding thresholds where water has enough energy to start eroding soil and form a channel head, for example the slope-area method which develop slope-area threshold relationships from the DEM, (Avcioglu et al., 2017; Imaizumi et al., 2010; Heine et al., 2004) or logistic regression (Heine et al., 2004). Such methods perform better in steeper terrain and poorer in flat terrain (McNamara et al., 2006). However, such algorithms depend on the natural formation of channel heads. In Sweden and other heavily drained countries, such as Finland and the Baltic States (Lõhmus et al., 2015) this has been offset by human influence and the numerous man-made channels draining the landscape. Also, based on the field data being point data of stream presence/absence and not actual channel heads it's not possible to use these methods. Instead the bias introduced by choosing a fixed threshold was investigated via a residual analysis.

The residual analysis (Fig. 3) show that there is a potential to further improve the channel networks by considering spatial variability, but the OPLS-DA could only explain 15% of overestimation and underestimation of stream channels (FPs and



FNs). For example, more FNs were found in the south west while FPs were found in the eastern and northern regions. While the coordinates themselves are not exerting controls on the channel head initiation, they capture many spatial gradients such as elevation, climate and even land use. Southern Sweden is characterized by an agricultural landscape while northern Sweden is mostly covered in forest. More FNs were found in south-western regions with high runoff during winter and on sedimentary

5    soils which could be a result of agricultural drainage that contributes to more stream channels. FP on the other hand were mostly found in drier regions towards the east where summer and spring runoff is lower as well as in northern Sweden. It's difficult to interpret these results since it's likely that both anthropogenic drainage systems for agriculture and forestry overlaps with quaternary deposits and topography. Additionally it's important not to read too much into these results since the residuals showed extensive overlap and the model had low predictive power.

Despite difficulties in modelling something so variable in space and time as small streams, the 2 ha stream channel network, while far from perfect, is still an improvement compared to the stream network on the property map (MCC increased from 0.39 to 0.46 (Table 1)), and therefore has many applications as a management tool. The 2 ha stream network can be used in forestry (Murphy et al., 2008a, 2008b) landscape planning and as best management practice as suggested

by (Kuglerová et al., 2017). In forestry it can be used to, for example, plan off road driving (Ågren et al., 2015; Mohtashami et al., 2017) or design machine free zones on sensitive soils in riparian areas along the channels (Kuglerová et al., 2014). Thereby preventing increasing loads of sediment or mercury to surface waters (Kreutzweiser and Capell, 2001). An important improvement compared to the property map is that it not only captures more headwater streams, the channels also follows the inundated channel in the DEM (Murphy et al., 2008a), placing the streams more correctly while streams on

the property maps are lines drawn approximately where the streams are. These new derived stream channel networks also form an integrated drainage network where the amount of water to each stream section can be calculated. During road construction this can be used to correctly place and dimension culverts, which can decrease the problem with roads washing out during flood events (Prasad et al., 2005). When such an event happens its costly not only for the land/road owner, it also comes at a cost for the downstream environments that receives increasing loads of sediment transport (Najafi and Bhattachar,

2011). Another benefit of the connectivity in the newly mapped streams, reflecting the connectivity of water, is that effects can be traced upstream or downstream. For example, any effect of a perturbation in the stream network at an upstream location (road construction, ditch cleaning, soil erosion, clear-cuts, ruts, etc.) can now be traced downstream. This can form an important planning tool in order to protect streams of high ecological status at a downstream site, as the quality there is not only reflecting the management of the downstream site but also the sum of all upstream management.

Another important use for the 2 ha stream network is for upscaling from process based studies to large scale budgets on any scale. Streams and rivers dominate the carbon dioxide emissions of inland waters and low order streams are suggested to be

disproportional contributors emitting more than 70 % of the total stream and river CO2 (Raymond et al., 2013). Wallin et al. (2018) calculated the evasion from streams in Sweden and estimated a total loss of CO2 to 2.7 Tg C yr$^{-1}$ and CH4 to 0.02 Tg C yr$^{-1}$. These numbers were derived from a 400,000 km stream network based on a 50 m x 50 m DEM. This stream length would correspond to >30 ha stream channel initiation threshold while the 2 ha network had a length of 2 639 163 km or 6.6 times longer. Clearly, this indicates that the CO2 and CH4 evasion, during most part of the year apart from extreme dry situations in Sweden is even larger that the suggested numbers from Wallin et al., (2018). This illustrates how important accurate stream networks are for both management and upscaling aquatic and climate research and constrain C emissions.

## 5 Conclusions

The majority of the smaller (<6 m width) stream channels in this study were missing on the Swedish property map. A more accurate stream network map has many uses in landscape planning and best management practice. Also, incorrect representations of stream networks, up to many times over, severely affect upscaling aquatic and climate research, a problem that increases with scale. This study clearly show that a solution to this is to map stream channel networks from high resolution digital elevation models. For the study region of Sweden, which was used as a test bench, a 2 ha flow initiation threshold yielded the optimum stream network, increasing MCC from 0.387 on the property map to 0.463. When applying the same methodology to other biomes it's necessary to adapt the models and find the optimum flow initiation threshold for that unique landscape.

### Data availability

The coordinates of the field plots are confidential information and will only be distributed with a non-disclosure agreement with the Swedish National Forest Inventory.

### Author contributions

AÅ conceived the study. AÅ and WL designed the study, performed the analysis and wrote the paper together.

### Competing interests

 The authors declare that they have no conflict of interest.





**Acknowledgements**

This project was financed by the EU InterReg Baltic Sea project WAMBAF, VINNOVA, Mistra - Future Forests, Formas - ForWater, and the Kempe foundation. Finally, we thank the National Inventory of Swedish Landscapes for providing the field reference plots.

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



Figure 1: The black points are field sites with observations of stream channels in the NILS database.



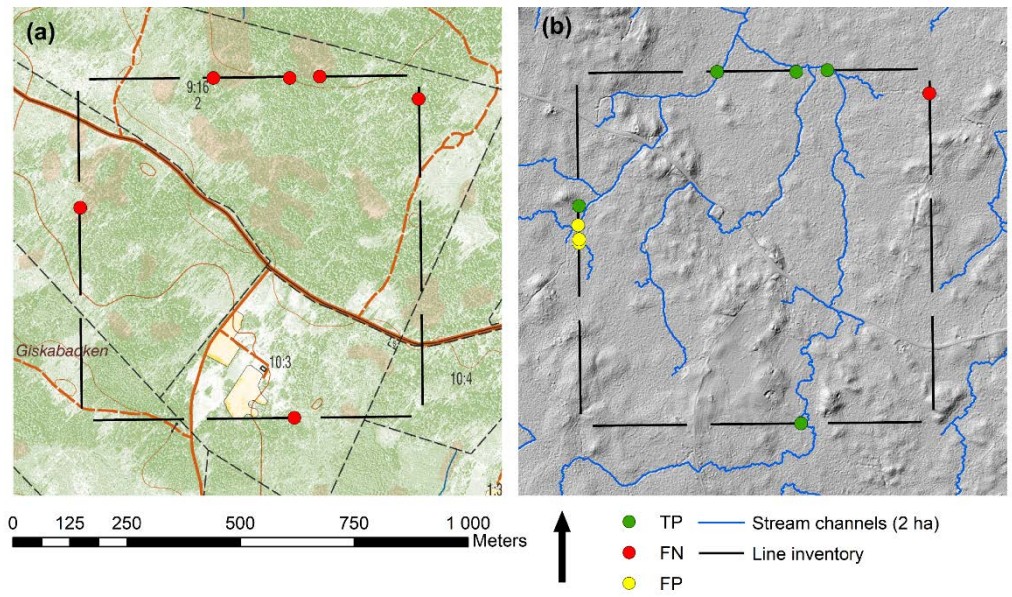

**Figure 2: Panel A shows an example of the 200 m segments in the line inventory as black lines superimposed on the Swedish property map (1:12 500) and the red points are field observations of stream channels, in this case all streams were missing on the map (FN). Panel B shows the same square but with a modelled stream channel network (2 ha) superimposed on a hillshade. The green points indicate streams that are accurately shown on the map (TP) while the red points indicate streams that are still missing on the map (FN).Yellow points are locations where the modelled stream channel network intersect an inventoried line without an observation of a stream channel (FP). To summarize the results in panel B; the 2 ha stream network captured 5 out of the 6 stream channels, misses one and generates 3 false positives in this case. This illustrates that the new DEM derived stream networks, while not perfect, is an improvement compared to the property map (panel A).**





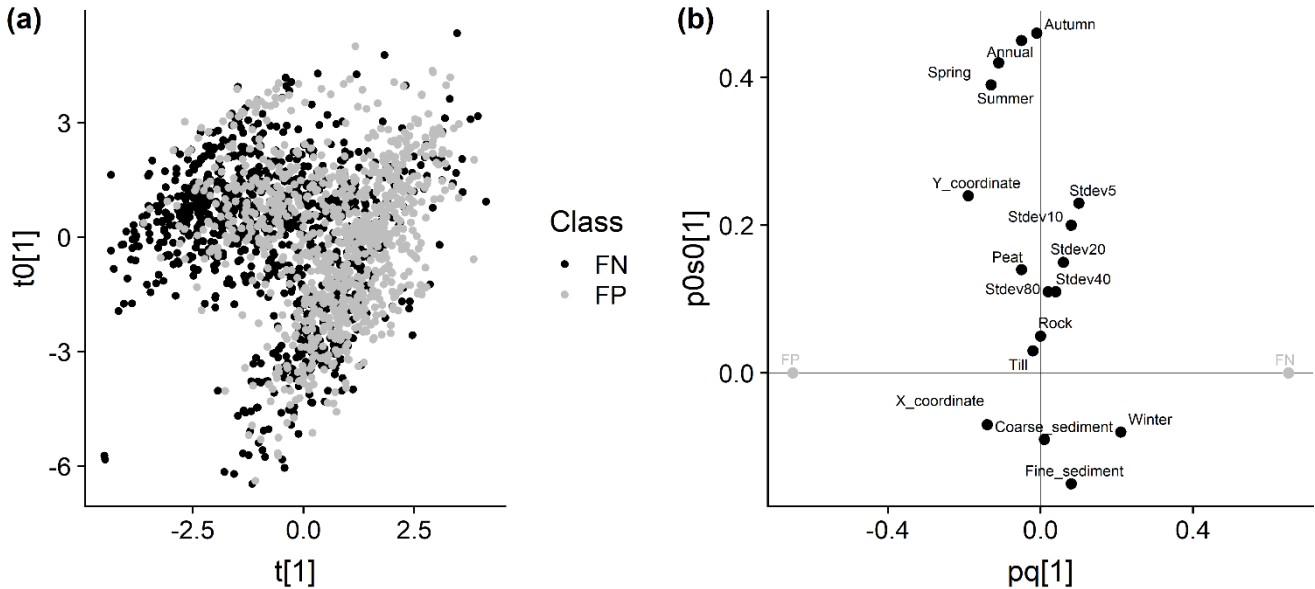

**Figure 3: A Score scatter plot from the OPLS-DA model. B) Loading scatter plot from the OPLS-DA model. The variables "Winter, Spring, Summer, Autumn and Annual" refers to the seasonal and annual runoff.**



**Table 1: Confusion matrix showing true positives (TP), true negatives (TN), false positives (FP) and false negatives (FN), as well as accuracy (ACC (%)), Mathews Correlation Coefficient (MCC) and stream length for stream channel networks with different f.i.t (flow initiation thresholds) which is the catchment area required to form a channel.**

| Mapped stream channels | TP | TN | FP | FN | ACC (%) | MCC | Length (km) |
|---|---|---|---|---|---|---|---|
| Channel network with 1 ha f.i.t | 2 185 | 611 575 | 5 076 | 931 | 99.0 | 0.456 | 3 811 247 |
| Channel network with 2 ha f.i.t | 1 869 | 613 360 | 3 291 | 1 247 | 99.3 | 0.463 | 2 639 163 |
| Channel network with 5 ha f.i.t | 1 429 | 614 845 | 1 806 | 1 687 | 99.4 | 0.447 | 1 603 704 |
| Channel network with 10 ha f.i.t | 1 107 | 615 513 | 1 138 | 2 009 | 99.5 | 0.416 | 1 089 122 |
| Channel network with 15 ha f.i.t | 914 | 615 793 | 858 | 2 202 | 99.5 | 0.387 | 864 370 |
| Channel network with 30 ha f.i.t | 684 | 616 103 | 548 | 2 432 | 99.5 | 0.347 | 576 452 |
| Property map stream lines | 618 | 616 301 | 143 | 2 705 | 99.5 | 0.387 | 593 042 |

