# Peer review of "The importance of better mapping of stream networks using high resolution digital elevation models – upscaling from watershed scale to regional and national scales"

_Hydrology and Earth System Sciences, 2019_

## Referee Comment (RC1) · Anonymous Referee #1 · 5 Feb 2019

Summary and Contribution

The authors determine the best stream initiation threshold to delineate headwater streams in Sweden by comparing modeled stream networks to a national inventory of the presence or absence of small stream channels. As the authors discuss, maps of headwaters are highly inaccurate, yet more realistic estimates of stream length are essential for countless applications related to watershed and natural resource management. While the objective of better headwater mapping is worthwhile, I believe the study is simplistic, especially considering the wealth of existing research that models

more accurate headwater stream networks over large areas.

General Comments

The authors suggest that this study is one of the first attempts to model headwater streams from DEMs over a large scale. Although the authors are correct that accurate headwater stream models over large areas remain a challenge, there are myriad studies that already use some derivation of DEM data to improve stream network mapping (e.g. Sun et al., 2011; Julian et al., 2012; Elmore et al., 2013; Russell et al., 2015; González-Ferreras & Barquín, 2017; Jensen et al., 2018; Jaeger et al., 2019). In addition, the practice of selecting a channel initiation threshold to match available field data – although useful for many practical applications – is quite basic and does not represent a significant contribution to the scientific literature. The authors state in the discussion that other methods are not appropriate for the presence/absence data from the national inventory, but I do not believe this statement is correct; on the contrary, presence/absence data are necessary for logistic regression models, for example.

The abstract mentions the selection of stream initiation thresholds across physiographic regions, but the authors appear to only create a single global model across all of Sweden. In addition, the residual analysis is quite limited. Although the description of the placement of false negatives and false positives in relation to quaternary deposits, terrain, and other factors is interesting, the authors could provide further analysis, visualization, and/or discussion to explain these patterns.

Finally, the manuscript requires further English language editing to correct spelling and grammar, particularly in the form of verb-number agreement and comma placement.

---

## Author Comment (AC1) · 26 Feb 2019

We would like to thank reviewer 1 for comments and suggestions for the manuscript. We believe that the comments have identified important areas which required improvement. We have taken the comments on board to improve and clarify the novelty manuscript.

The scope of HESS is to encourage and support both fundamental and applied research. This study does not advance the fundamental research, but, it fills an important research gap in applied science. Which is to generate maps that can be used in practice, based on the best available science. It's it true that there are studies that has helped researchers understand the controls on stream head formations and perennial/non-perennial headwaters across larger regions. Jaeger et al. (2019) that the referee mentions, is an excellent example of such a study. However, such regional studies often aggregate the DEMs to larger pixels to minimize computational resources. Jaeger et al. (2019) use 30 m pixels, González-Ferreras & Barguín (2017) use 25 m, Julian et al. (2012) and Elmore et al. (2013) use 10 m. While this resolution is good for advancing the processes understanding it's important to work on a higher resolution when generating maps of headwaters that can be used in practice and that accurately places the modelled streams. In our case all field mapped streams had a width less than 6 m. For mapping purposes it therefore makes sense to use a scale <6 m. Russell et al (2015) worked on a 6.1 m resolution, stating that "The resolution of 6.1 m (20 ft) was selected to maximize the accuracy of terrain derivatives while minimizing computational resources." Jensen et al. (2018) use a resolution of 3 m. These studies are examples of better resolutions for mapping small streams, however that also meant that they worked on a catchment scale. Jensen worked on 3 catchments smaller than 45 ha and Russell worked on 7 catchments ranging 1.1-4.8 km2. In Lidberg et al. (2017) we showed that the accuracy of topographically derived stream networks increased with increasing DEM resolution. DEM derived stream networks were field validated against >30 000 field observations of stream/road intersections from 9 catchments ranging 68-1 993km2. Stream networks from the 2 m DEM intersected roughly twice the number of stream/road intersections as stream networks from the 16 m DEM. We also found that working with high resolution DEMs (2 m) requires special pre-processing methods as new challenges are introduced with increasing resolution because the effect of anthropogenic artefacts such as road embankments and bridges. These are problematic during the pre-processing step because they are elevated above the surrounding landscape and act as artificial dams (Figure 1). The effect of different pre-processing methods were studied in Lidberg et al (2017) and that knowledge was applied in this
**study.**

This study is the first study where all headwater stream channels have been mapped throughout an entire country (Sweden is 450 295 km2) at a high resolution (2 m). A drawback of working in a 2 m resolution on this large scale is the processing times for the modelling of stream channels. But, the rewards are that the modelled streams on the digital maps mostly follow the inundations of the small streams and ditches that can often be picked up by really high-resolution DEMs (Figure 2). We have now more specifically expressed the novelty of this study in the manuscript and also added a new figure in the manuscript, highlighting the benefits of working on a high resolution when mapping headwaters.

The development of LiDAR models have also changed forest management. It's now possible to segmenting individual trees from the LiDAR point clouds (Li et al. 2013) and to estimate tree height (Suárez et al. 2005) and biomass (Popescu 2007) of individual trees. Such models can be used to assess the vegetation structure (Morsdorf et al. 2004) and forest succession (Falkowski et al. 2009) and be used in, for example, fire risk assessment and fire behavior modeling as well as wildlife habitat assessment. As forest management work on higher resolutions (individual trees) in order to optimize different goal such as production or biodiversity, it's also important to develop tools for surface water protection that meets the demands of high-resolution maps by the future forest industry. Better mapping of headwater streams is a key part of this work.

When it comes to binary data in models of headwaters, studies like González-Ferreras & Barquín, (2017) and Jaeger et al. (2019) use machine learning (Random Forest) to model perennial/non-perennial streams. We explored the possibility of using machine learning algorithms in our study. However, our dataset did not contain stream heads and further it is hugely imbalanced with 3 323 streams and 616 444 non-streams.

To create a balanced dataset that could be analyzed further in the residual analysis we used the following approach: "Prior to analysis a balanced dataset was created by
randomly selecting and equal number of FN and FP and all variables were scaled and centered."

As pointed out by the reviewer, logistic regression is a common method for classifications. However, all regression methods are based on a number of fundamental assumptions that needs to be met. Our dataset met all the assumptions for a logistic regression – except one, it violates the assumption of no multicollinearity. Meaning that several of our independent variables are highly correlated with each other. Therefore we cannot use a regression method. A way of dealing with multicollinearity is to use a multivariate approach where principal components are extracted from the dataset of X- variables. These component are extracted so they are orthogonal to each other, meaning that the components are not covariate. That is why the OPLS-DA was the correct statistical method to use on our dataset. In the article we wrote "First the Xmatrix was tested for multicolinearity, using IBM SPSS Statistics 24 and since many of the explanatory variables in the residual analysis showed multicolinearity a multivariate approach was used."

Having said that, we are also aware that the regression models often are a robust method, and we also tested a Logistic Regression using IBM SPSS Statistics v. 24. The Hosmer and Lemeshow test for significance showed that only three of the models were significant, step 2, 4 and 5. R2 for those models range 0.14-0.21, depending on the chosen method for R2 (Cox & Snell R2 or Nagelkerke R2). That's of the same order as the OPLS-DA which had R2Y(cum) = 0.16 and Q2(cum) = 0.15. The important variables in step 5 were Winter, StDev5, StDev10, X-coordinates and fine sediments. Similar to the findings in the OPLS-DA. So, applying a logistic regression does not change the major findings of this study, nor does it improve the explanatory power of the model (of the same order). However, bear in mind that it's also statistically wrong to use a regression method on this dataset.

It's true that we only create a single global model across all of Sweden and then use the residual analysis to infer how the models can be improved in the future by taking into
account landscape variability. We have now expanded the discussion on the residual analysis in the manuscript. We believe that in order to further improve the modelling and adjusting the stream initiation thresholds across physiographic regions we need to create an entire new field dataset with stream heads mapped throughout the nation. While it's our aim to work towards that goal during the upcoming years, we have still learned a lot in this study which fills an important gap in the applied science community. We show that it's possible to generate stream networks on a high resolution (2 m) over an entire country. The plan is that the digital stream network (2 ha) will be made available through the Geodata Collaboration which give authorities, county councils / regions, municipalities, public-sector organizations and forest owners access to the maps. Wide access to these maps will improve the basis for landscape planning, including forestry, and can contribute to reduce the export of nutrients and hazardous substances to streams, lakes and the Baltic Sea.

Regarding the comments on English spelling and grammar errors. We will send the manuscript on languish editing to address these problems.

References

Elmore, A. J., J. P. Julian, S. M. Guinn, and M. C. Fitzpatrick. 2013. Potential Stream Density in Mid-Atlantic U.S. Watersheds. PLoS ONE. doi:10.1371/journal.pone.0074819.

Falkowski, M. J., J. S. Evans, S. Martinuzzi, P. E. Gessler, and A. T. Hudak. 2009. Characterizing forest succession with lidar data: An evaluation for the Inland Northwest, USA. Remote Sensing of Environment. doi:10.1016/j.rse.2009.01.003.

González-Ferreras, A. M., and J. Barquín. 2017. Mapping the temporary and perennial character of whole river networks. Water Resources Research. doi:10.1002/2017WR020390.

Jaeger, K. L., R. Sando, R. R. McShane, J. B. Dunham, D. P. Hockman-Wert, K. E.
Kaiser, K. Hafen, J. C. Risley, et al. 2019. Probability of Streamflow Permanence Model (PROSPER): A spatially continuous model of annual streamflow permanence throughout the Pacific Northwest. Journal of Hydrology X 2. The Authors: 100005. doi:10.1016/j.hydroa.2018.100005.

Jensen, C. K., K. J. McGuire, and P. S. Prince. 2017. Headwater stream length dynamics across four physiographic provinces of the Appalachian Highlands. Hydrological Processes 31: 3350–3363. doi:10.1002/hyp.11259.

Julian, J. P., A. J. Elmore, and S. M. Guinn. 2012. Channel head locations in forested watersheds across the mid-Atlantic United States: A physiographic analysis. Geomorphology. doi:10.1016/j.geomorph.2012.07.029.

Li, W., Q. Guo, M. K. Jakubowski, and M. Kelly. 2013. A New Method for Segmenting Individual Trees from the Lidar Point Cloud. Photogrammetric Engineering & Remote Sensing. doi:10.14358/pers.78.1.75.

Lidberg, W., M. Nilsson, T. Lundmark, and A. M. Ågren. 2017. Evaluating preprocessing methods of digital elevation models for hydrological modelling. Hydrological Processes 31: 4660–4668. doi:10.1002/hyp.11385.

Morsdorf, F., E. Meier, B. Kötz, K. I. Itten, M. Dobbertin, and B. Allgöwer. 2004. LIDAR-based geometric reconstruction of boreal type forest stands at single tree level for forest and wildland fire management. In Remote Sensing of Environment. doi:10.1016/j.rse.2004.05.013.

Popescu, S. C. 2007. Estimating biomass of individual pine trees using airborne lidar. Biomass and Bioenergy. doi:10.1016/j.biombioe.2007.06.022.

Russell, P. P., S. M. Gale, B. Muñoz, J. R. Dorney, and M. J. Rubino. 2015. A spatially explicit model for mapping headwater streams. Journal of the American Water Resources Association. doi:10.1111/jawr.12250.

Suárez, J. C., C. Ontiveros, S. Smith, and S. Snape. 2005. Use of airborne LiDAR and
aerial photography in the estimation of individual tree heights in forestry. Computers and Geosciences. doi:10.1016/j.cageo.2004.09.015.

---

## Referee Comment (RC2) · Anonymous Referee #2 · 6 May 2019

The authors used a novel national dataset of stream points and compared the location and total number of stream points with those on a property map and modeled stream maps based on a lidar based digital elevation model using different stream initiation thresholds (thresholds in accumulated area). This work is important because most stream maps under-represent the total stream length; the use of maps that do not depict the stream network correctly leads to large errors when upscaling riparian length or $CO_2$ evasion from streams. However, very few results are shown in the manuscript (just one example map) and there is almost no discussion of regional differences in the optimal stream initiation threshold. As such the manuscript doesn't reach its full potential and doesn't highlight the novelty of the study well. Also, the discussion remains largely limited to comparisons with a more detailed study in the Krycklan catchment, while other (international) studies could be mentioned here as well. Furthermore, there is no discussion on how the DEM pre-processing steps affect the results. The manuscript contains relatively many typos and some awkward phrases and would have benefitted from a careful round of editing before submission. For example, with 'headwaters' sometimes the catchment and other times the streams are meant. Similarly, it is not always clear for 'streams' if the simulated streams based on the DEM, the streams from the property map or the stream points from the NILS dataset are meant. This is confusing and could easily be solved with clearer writing. Some of these are highlighted in the attached pdf.

Specific comments:

Abstract:

P1L16: Mention the Nils dataset in the abstract. Now it is not clear from the abstract to what dataset you compared your modeled stream networks.

P1L21: You don't really illustrate the effect of the stream network on upscaling aquatic research or climate research but rather your results show that these upscaling results depend on which stream network are used. A bit more careful wording is needed here. Also, what is meant with 'upscaling climate research'?

Introduction:

References to studies that compared stream networks on maps and actual stream networks seem to be largely missing. Because this is exactly the topic of this paper, this is strange and makes it more difficult to judge the novelty of this research. For example: • Brooks RT, Colburn EA. 2011. Extent and Channel Morphology of Unmapped Headwater Stream Segments of the Quabbin Watershed, Massachusetts1.

JAWRA Journal of the American Water Resources Association, 47: 158-168. DOI: 10.1111/j.1752-1688.2010.00499.x. • Russell PP, Gale SM, Muñoz B, Dorney JR, Rubino MJ. 2015. A Spatially Explicit Model for Mapping Headwater Streams. JAWRA Journal of the American Water Resources Association, 51: 226-239. DOI: 10.1111/jawr.12250. • Fritz, K. M., Hagenbuch, E., D'Amico, E., Reif, M., Wigington, P. J., Leibowitz, S. G., Comeleo, R. L., Ebersole, J. L., and Nadeau, T.-L.: Comparing the Extent and Permanence of Headwater Streams From Two Field Surveys to Values From Hydrographic Databases and Maps, JAWRA Journal of the American Water Resources Association, 49, 867-882, 10.1111/jawr.12040, 2013.

Methods:

P4L8-10: More information needs to be given for the methods used to create the NILS database. This is the data to which your modeled stream networks are compared. It appears to be a very novel dataset that makes this work novel but almost no information is given on it (and what is written on it is not clear until one looks at figure 2).

P4L14: Quantify this movement and snapping of the streams. For what fraction of the sites was this the case? To me the 20 m distance seems a lot considering the 200 m line segments, particularly for flat areas where streams are not incised. Please comment on the effect of this step here (and in the discussion!).

P4L29/P514: Is the property map really the most logical map to look for streams? Add more information on how this map was created.

P4L29: Didn't you use any smoothing or the filling of the DEM before creating the stream networks based on the D8 method? Also, I have the feeling that there is a bit of a circular work flow here. First you use the stream map to adjust the DEM (burn in the streams), then you use the adjusted DEM to simulate where the streams are, and finally you compare the simulated stream network with the original map that you used for burning the streams into the DEM. Doesn't this burning of the stream network affect the simulated stream network and especially the number of FPs and FNs? This is not

mentioned nor discussed anywhere.

P5L6: Please describe why you used the D8 method and not the DMinf method. Generally the D8 method leads to many "parallel streams" in headwater catchments and this would significantly affect the number of streams. Please justify the choice of the D8 method and discuss this in more detail.

P6L2: Some background information on the statistical methods used would be useful. Not everyone is familiar with OPLS-DA.

Results:

Almost no results are shown in the results section. Figure 2 is nice but it would be interesting to show this for more locations (e.g., a site in the north and one in the south, a flat site and steep site, etc.). Also maps with the NILS sites and the fraction of correctly mapped stream points, FP and FN for these points would be very useful. Now the results are very thin. I think that the work is interesting and that it uses is a unique database but none of the regional variability in the modeled stream networks nor the goodness of fit of the simulated networks is shown. This information needs to be shown to understand the residual analysis. Almost more results are described in the discussion section than in the results section.

Discussion:

The discussion is very limited and focused on comparisons with a previous study of the authors in the Krycklan catchment. Other studies could and should be referenced in the comparison of the modeled and simulated stream networks as well (see comment for the introduction). Also, I would have expected some discussion of the trade off between the number of FPs and FNs but it seems that the authors advocate optimizing the FP. Is this really the best from a management point of view? or would too many FNs mean that the maps won't be used at all. Some discussion of this trade off would be useful.

It would also be useful if there was more discussion on the regional patterns in the goodness of fit of the simulated networks. I realize that the focus is on finding the national scale optimal stream network initiation threshold value but there must be large regional differences that are interesting to explore. Not knowing what these regional differences are makes it hard to appreciate the value of having a national average stream initiation threshold value.

Finally there is almost no discussion of the effects of the uncertainties caused by the methods or the DEM pre-processing steps, e.g. the burning of the stream network or the snapping of the streams. P9L16-29: The down-valley changes in topography are important as well. See: Prancevic, J. P., and Kirchner, J. W.: Topographic Controls on the Extension and Retraction of Flowing Streams, Geophysical Research Letters, 46, 2084-2092, 10.1029/2018gl081799, 2019.

P10L21-29: I find this a bit of a long stretch. Wouldn't you use for something local (and expensive) like building a road local knowledge and field observations or more detailed maps than a national scale stream map derived from a national average stream area initiation threshold? I can see the advantage of a national scale map for up-scaling biogeochemical fluxes or the size of riparian corridors or many other things but not for local road construction.

P11L4: But this doesn't take all the FNs into account and thus uses the wrong stream length/points as well. Shouldn't you at least mention that this would cause an over-estimation of the total length? Admittedly this effect is small compared to the huge uncertainty due to using the stream map with far too few streams but it should at least be acknowledged.

Conclusion:

The conclusion doesn't highlight the novelty of the research well.

Please also note the supplement to this comment:

https://www.hydrol-earth-syst-sci-discuss.net/hess-2019-34/hess-2019-34-RC2-supplement.pdf

**Supplement:**

[revised manuscript text omitted]

---

## Referee Comment (RC3) · Anonymous Referee #3 · 6 May 2019

This paper is probably one of the worst I have ever been asked to review. The only interesting information I got by reading it was to discover the NILS dataset, which was not collected by the authors. My recommendation is to reject this paper.

1. Formal problems:

- Poor English, with a lot of typos, grammatical errors, improper expressions in written English ("it's"), and some very awkward sentences, so awkward that the reader loses the sense.

[Figure]

- The authors use terms and abbreviations that are not canonical in environment science, without proper explanation, e.g., "breaching", Matthews Correlation Coefficient, Residuals (used in a very odd way in this paper), "local topography was calculated as a standard deviation of the digital elevation model in a moving window" (p6, L27), IBM SPSS Statistics 24, OPLS-DA, SIMCA 14, R2Y(cum), R2X(cum), Q2(cum), loading scatter plot.

- Even "stream burning" would deserve a short explanation although classically used for a long time in DEM processing.

- The methods are very hard to follow, because of the above deficiencies, and also because the authors have no sense of structure. In particular Fig2 should not be called in the Results section but in Material and method, to illustrate the NILS dataset. This dataset, which seems to be extremely rich and interesting would deserve much more explanations. The beginning of Section 3 (p7, L4-9) gives results on the field investigation (NILS) and should go in the dedicated section (2.2). The Swedish property map should be introduced independently from the modelled stream networks as it serves as an (independent?) reference.

- The Results and Discussion sections also suffer structural problems, with some results in the Discussion (in particular the first paragraph with %, the links of which with Table 1 is not straightforward), some elements of the Discussion which repeat the Introduction, and some other elements which open future perspectives and would rather fit in the Conclusion.

2. Scientific problems:

- No clear scientific questions. The ones I gathered are at the end of the Introduction (p3, L11-12, with typo): "A research question that remains is therfore; How does one select the correct stream initiation threshold and validate DEM derived stream networks when scaling up from a catchment scale to regional or national scale? [. . .] Therefore the aim of this study is to determine the optimum threshold for stream initiation on a

national scale, using the Swedish landscape at a test bench." But both questions are ill-posed since the paper is not about upscaling (the study only addresses headwater catchments in 25-km$^2$ squares) and looking for "the" optimal threshold for the large scale is extremely naive given recent results showing that the initiation threshold is spatially variable, and depends on several factors, like geology, land use, or climate (e.g. Colombo et al., 2007; Luo et al., 2016; Schneider et al. 2017). The authors seem to be aware of the problem, cf. the last line of the paper ("When applying the same methodology to other biomes it's necessary to adapt the models and find the optimum flow initiation threshold for that unique landscape"), so what's the point of the paper?

- Only two pieces of result! They are limited to one table and one figure with a totally insufficient caption (Fig. 3), so the reader does not even know what he/she looks at. This is far from being enough, and all the more as these elements are not properly commented.

- Circular use of the property map, which serves as a benchmark against which the modelled stream networks compare favorably, but on the other hand, the corresponding stream lines were "burnt" in the DEM used to model the steam networks with several initiation thresholds.

- Many unsupported conclusions, especially in the Discussion (subjective or speculative opinions should appear as such), and some abusive conclusions. In particular, the accuracy values in Table 1 are all very close to each other and should thus be used with caution. In contrast, the Matthews Correlation Coefficients are never very good (always lower than 0.5 which is not outstanding for a correlation coefficient) so it is misleading to conclude that "a 2 ha flow initiation threshold yielded the optimum stream network.

- The discussion misses important points, notably the influence on the results of (i) the LiDAR DEM quality, (ii) the imprecision of the stream location, (iii) the nature of the field survey which does not investigate the full squares but only their borders, (iv) the

particular geology and geomorphology of Sweden and consequences for generalizing the results.

- The title is not explicit, since it focuses on upscaling, which is absolutely not the key point of the study.

3. Cited references:

Colombo, R., J. V. Vogt, P. Soille, M. L. Paracchini, and A. de Jager (2007), Deriving river networks and catchments at the European scale from medium resolution digital elevation data, Catena, 70(3), 296–305.

Luo, W., J. Jasiewicz, T. Stepinski, J. Wang, C. Xu, and X. Cang (2016), Spatial association between dissection density and environmental factors over the entire conterminous United States, Geophys. Res. Lett., 43, 692–700, doi:10.1002/2015GL066941.

Schneider A.S., Jost A., Coulon C., Silvestre M., Théry S., Ducharne A. (2017). Global scale river network extraction based on high-resolution topography, constrained by lithology, climate, slope, and observed drainage density. GRL, 44, 2773–2781.

---

## Editor Comment (EC1) · Bettina Schaefli (Editor) · 20 Nov 2019

Based on the outcome of the public discussion, the authors decided to withdraw the paper and to resubmit a new version in due time. Please check the HESS/HESSD database for the new version.